# Specific Application Features of Ti-TiN-(Ti,Cr,Al)N, Zr-ZrN-(Zr,Mo,Al)N, and ZrHf-(Zr,Hf)N-(Zr,Hf,Cr,Mo,Al)N Multilayered Nanocomposite Coatings in End Milling of the Inconel 718 Nickel-Chromium Alloy

**Alexey Vereschaka** [1,*], **Filipp Milovich** [2], **Nikolay Andreev** [2], **Mars Migranov** [3], **Islam Alexandrov** [1], **Alexander Muranov** [1], **Maxim Mikhailov** [1] **and Aslan Tatarkanov** [1]

1 Institute of Design and Technological Informatics of the Russian Academy of Sciences (IDTI RAS), 127994 Moscow, Russia
2 Materials Science and Metallurgy Shared Use Research and Development Center, National University of Science and Technology MISiS, 119049 Moscow, Russia
3 VTO Department, Moscow State Technological University STANKIN, 127055 Moscow, Russia
* Correspondence: dr.a.veres@yandex.ru

**Abstract:** This article discusses the specific application features of end mills with Ti-TiN-(Ti,Cr,Al)N, Zr-ZrN-(Zr,Mo,Al)N, and ZrHf-(Zr,Hf)N-(Zr,Hf,Cr,Mo,Al)N multilayer nanocomposite coatings during the machining of the Inconel 718 nickel–chromium alloy. The hardness, fracture resistance during scratch testing, structure, and phase composition of the coatings were studied. The tribological properties of the samples were compared at temperatures of 400–900 °C. Tests were conducted to study the wear resistance of the coated end mills during the milling of the Inconel 718 alloy. The wear mechanism of the end mills was studied. It was found that in comparison with the other coatings, the Zr-ZrN-(Zr,Mo,Al)N coating had the highest hardness and the lowest value of the adhesion component of the coefficient of friction at high temperatures. However, the Zr-ZrN-(Zr,Mo,Al)N coating exhibited good resistance to cracking and oxidation during the milling of the Inconel 718 alloy. Based on the above, the Zr-ZrN-(Zr,Mo,Al)N coating can be considered a good choice as a wear-resistant coating for the end milling of the Inconel 718 alloy.

**Keywords:** Inconel 718 alloy; multilayer nanocomposite coatings; wear resistance; end milling; wear mechanism

## 1. Introduction

Inconel 718 is a heat-resistant alloy based on a nickel–chromium–iron system that combines resistance to corrosion and high strength [1,2]. This alloy also demonstrates high resistance to creep (deformation) at temperatures up to 700 °C. Inconel 718 can be effectively used to manufacture products with increased elasticity (for example, springs). The Inconel 718 alloy is specially designed for high-temperature applications and is widely used. In the United States, this alloy accounts for almost 50% of the total production of industrial alloys with increased thermal stability [2]. Inconel 718 is one of the most widely used alloys from the Inconel group. The chemical composition of Inconel 718 includes nickel, chromium, and iron, with the addition of aluminum, titanium, molybdenum, and niobium (see Table 1).

Inconel 718 is characterized not only by high strength and resistance to corrosion, but also by good machinability and weldability. It is also worth noting such advantages of the Inconel 718 alloy as its resistance to corrosion at welds and high tensile strength at temperatures up to 700 °C. According to the NACE MR0175 standard [3], Inconel 718 is type 4d, and thus it can be used in conditions of exposure to various combinations of chlorides and to hydrogen sulfide. The Inconel 718 alloy was designed primarily as a

special material for the bodies of supersonic aircrafts. At present, Inconel 718 is actively used for the manufacture of parts for gas turbines, elements of rocket and aircraft engines (for example, compressor blades), and spacecrafts. The Inconel 718 alloy is also used in the manufacture of nuclear reactors and various products for the chemical and oil and gas industries [1,2,4].

**Table 1.** Chemical composition of the Inconel 718 alloy [1].

| Ni | Fe | Cr | Cu | Mo | Nb | C | Mn | Si | Ti | Al | Co |
|---|---|---|---|---|---|---|---|---|---|---|---|
| 50.00–55.00 | Remainder | 17.00–21.00 | 0.30 max | 2.80–3.30 | 4.75–5.50 | 0.08 max | 0.35 max | 0.35 max | 0.65–1.15 | 0.20–0.80 | 1.00 max |

Despite the significant advantages described above, the Inconel 718 alloy can be considered a difficult-to-cut material [5–7]. Due to the considerably low thermal conductivity of Inconel 718, the cutting zone can be heated to high temperatures. Another problem in the machining of the Inconel 718 alloy is its increased work hardening. The structure of Inconel 718 can contain highly abrasive carbide grains. Finally, this alloy is characterized by considerably high hardness (about 330 MPa [8]) and a high tendency to abrasion and interdiffusion when using carbide tools. Due to the above reasons, Inconel 718 is machined at low cutting speeds, which reduces the machining efficiency and increases the production costs [6,7].

It is possible to increase the cutting speed and, consequently, the machining efficiency with the use of tool materials with enhanced-properties coated materials. Coatings with rationally selected composition and structure can reduce the adhesion with the material being machined and, accordingly, allow decreasing the temperature in the cutting zone [9,10]. Moreover, coatings can reduce interdiffusion [11–13] and increase the resistance to abrasive wear [14,15].

In general, during the machining of Inconel 718, the coatings of TiN, (Ti,Al)N, and (Ti,Cr,Al)N are used [16–22]. For these coatings, good hardness and wear resistance, combined with considerable heat resistance, are typical. Meanwhile, coatings of different compositions can further reduce the adhesive interaction with the material being machined. At elevated temperatures, typical for the conditions in which Inconel 718 is machined, the oxidation and diffusion mechanisms of wear are more active, and this fact serves as an additional motivation for studying coatings with alternative compositions. Accordingly, coatings are expected to have good resistance to the above factors.

The milling process is accompanied with a high level of alternating mechanical and thermal loads acting on the cutting tools. Thus, the coatings used in milling should also resist brittle fracture and cracking [23,24]. The coatings of TiN and, especially, (Ti,Al)N and (Ti,Cr,Al)N are quite hard but also very brittle [25–27], which largely limits their use in intermittent cutting conditions.

The coatings based on the systems of ZrN, CrN [28,29] and their combinations with the introduction of additional elements can become an alternative to the coatings based on the system of TiN. Hafnium (Hf) introduced into the coating composition can provide an increase in thermal stability [30]. Another advantage of hafnium nitride is the higher values of such parameters as Young's modulus and bulk modulus in comparison with the systems of TiN and ZrN [30]. In [31], the authors also noted a good resistance to oxidation of the HfN system. The best combination of properties is obtained when Hf is introduced into the systems based on ZrN, CrN, and TiN. For example, the coating of (Zr,Hf)N demonstrated the formation of tribologically active oxide films of $ZrO_2$ and $HfO_2$, which prevented further oxidation [32]. In comparison with the coating of ZrN, the coating of (Zr,Hf)N is characterized by noticeably higher wear resistance and adhesion strength to the substrate [33]. The introduction of Zr and Hf into the composition of the TiN coating provided a noticeable increase in thermal stability and resistance to oxidation [34]. In multicomponent coatings (in particular, the (Hf,Nb,Ti,V,Zr)N coating [35,36]), hafnium

provided a reduction in the defectiveness due to temperature diffusion and a slowdown in the spinodal decomposition at the boundaries of the coating layers and grains. Another positive effect from the introduction of hafnium into the composition of multicomponent coatings is associated with the formation of saturated nitride phases and the strengthening of the substitution solid solution [37]. In addition, an important advantage associated with the presence of Hf and Zr in multicomponent coatings is related to the positive transformation of their tribological properties at elevated temperatures due to the formation of $ZrO_2$ and $HfO_2$ oxide films [38,39].

Molybdenum (Mo) is another element that has a favorable effect on a coating's properties. In particular, when the (Cr,Mo)N coating is heated up to the temperature of 700 °C, a protective tribological film of $MoO_3$ is formed [40,41]. Like hafnium, molybdenum has a positive effect on the properties of multicomponent coatings [42–44]. In particular, the coatings consisting of (Al,Cr,Mo,Si,Ti)N [42] and (Cr,Ta,Nb,Mo,V)N [43] demonstrated high hardness and wear resistance due to solid solution strengthening and the formation of a strong "metal–nitrogen" bond.

One more important element influencing the properties of nitride coatings is aluminum (Al). The introduction of Al into the coating composition makes it possible to simultaneously enhance such coating properties as hardness and resistance to wear and heat [45–47]. Another positive effect arises from the formation of a dense tribologically active oxide film of $Al_2O_3$, providing a noticeable increase in the performance properties of the coatings [48].

The properties of coatings can be enhanced not only by achieving optimal compositions, but also by varying the parameters of their architecture. Multilayered coatings and coatings with a nanolayered structure are actively used [49–51].

Given the considerably tough conditions of the end milling of titanium alloys, in particular, Inconel 718, the coating should simultaneously be characterized by high resistance to wear, thermal stability, and resistance to oxidation. In case of intermittent cutting conditions, the resistance to cracking is also an important parameter. Thus, the following coatings were selected:

1.  Coating ZrHf-(Zr,Hf)N-(Zr,Hf,Cr,Mo,Al)N. This coating is based on the ZrN system, providing good resistance to cracking. The introduction into the coating composition of such elements as hafnium (Hf), molybdenum (Mo), and aluminum (Al) can enhance thermal stability and resistance to oxidation wear due to the formation of hard protective films. At the same time, Cr and Al can provide high hardness and wear resistance in combination with heat resistance. The architecture of the coating consists of three functional layers [52–54]:

    - an adhesive layer of ZrHf, 30–50 nm thick, which provides not only good adhesion to the substrate, but also a leveling effect due to the plastic filling of micropores and irregularities on the substrate surface,
    - a transition layer of (Zr,Hf)N, 0.7–1.0 μm, which provides a smoothing transition of the substrate properties to the wear-resistant layer,
    - a wear-resistant layer of (Zr,Hf,Cr,Mo,Al)N, 3.0–3.5 μm which, in turn, has a nanolayer structure.

2.  Coating Zr-ZrN-(Zr,Mo,Al)N, which is similar in its composition to the ZrHf-(Zr,Hf)N-(Zr,Hf,Cr,Mo,Al)N coating, but contains no Hf or Cr.

3.  The properties of the considered coatings were compared with the properties of the coating consisting of Ti-TiN-(Ti,Cr,Al)N with similar parameters, including total thickness, thickness of the functional layers, and modulation period in the wear-resistant layer. This coating was chosen as an object of comparison due to the widespread use of coatings based on the (Ti,Al)N and (Ti,Cr,Al)N systems during the machining (in particular, the end milling) of titanium alloys [16–22].

In the following, to simplify the description, the coatings will be designated using the composition of their wear-resistant layer—(Zr,Hf,Cr,Mo,Al)N, (Zr,Mo,Al)N, or (Ti,Cr,Al)N,

respectively—with the implication that all the considered coatings have the three-layer architecture described above. In the cases where it is important to emphasize the three-layer architecture of the coatings, their full designation will be used.

## 2. Materials and Methods

The coatings under consideration were deposited on carbide milling cutters and indenters using the VIT-2 unit (IDTI RAS–MSTU STANKIN, Moscow, Russia) [55–61]. Three evaporators in different systems were used during the deposition process. An Al (99.8%) cathode was installed on an evaporator of the filtered cathodic vacuum arc deposition (FC-VAD) system [62,63], which was associated with a high level of formation of microparticles during the sputtering of aluminum and the ability of the FCVAD system to separate up to 98% of such microparticles from the plasma flow. Cathodes of Zr-Hf (49.8/50.3%) and Cr-Mo (47.6/52.3%), as well as Ti (99.6%) and Cr (99.9%) (depending on the type of coating being deposited) were installed on the evaporators of the Controlled Accelerated Arc (CAA-PVD) system [64,65], which also contributed to a decrease in the amount of microparticles, with a noticeable reduction in energy consumption. Before being placed in the chamber, the samples underwent a preliminary preparation, including washing in a special solution with ultrasonic stimulation, rinsing in purified water, and drying in a stream of heated purified air. The samples were placed on a special toolset, installed on a turntable that provided planetary rotation, thus not only ensuring the uniformity and homogeneity of the coatings being deposited, but also forming the nanolayer structure of the coating. Before the coating deposition, the samples placed in the chamber underwent the procedure of cleaning and thermal activation in a gas and metal plasma flow in the following conditions: gas (Ar) pressure 2.0 Pa, voltage on the substrate 110 V. The functional layers of the coatings were deposited during the following time intervals: the deposition of the adhesive layer took 3 min, that of the transition layer took 15 min, and that of the wear-resistant layer took 45 min. During the processes of cleaning/activation and deposition of the coatings, the turntable rotation speed was n = 0.7 rpm.

During the deposition of the coatings, the cathode arc current was 110 A, 140 A, 160 A, 110 A, and 75 A, respectively, for the cathodes of Zr-Hf, Cr-Mo, Al, Ti, and Cr at a voltage on the substrate of $-150$ V.

At various stages of the coating deposition, the following nitrogen pressure ($p_N$) in the chamber was maintained:

Pumping and heating of the vacuum chamber: $p_N$ = 0.06 Pa
Heating and cleaning of the products with gaseous plasma: $p_N$ = 2.00 Pa
Deposition of the coating: $p_N$ = 0.42 Pa
Cooling of the products: $p_N$ = 0.06 Pa
The surface temperature of the samples was 650–700 °C.

The wear resistance of the coatings was compared when the coated samples of the Inconel 718 alloy were subjected to end milling on a Knuth WF 4.1 vertical milling machine (Knuth Werkzeugmaschinen GmbH, Wasbek, Germany). End milling cutters with the diameter of 12 mm were used, with the total length of 78 m and the working length of 26 mm (Figure 1). The tool material was carbide of H10F (WC + 10% Co). The angle of groove inclination was 35°. End milling cutters with the considered coatings and without coatings were used. Five tests were carried out for each type of coating, after which the average value of wear on the flank face was determined [66,67].

The following cutting conditions were used: n = 1000 rpm, f = 60 mm/min, $a_p$ = 1 mm, b = 4 mm. The flank face $VB_{max}$ = 0.3 mm was assumed as a wear criterion.

A mechanical tester of CB-500 (Nanovea, Irvine, California, USA) with a nanomodule was used to measure hardness and elastic modulus. This tester was equipped with a precision piezo driver and a highly sensitive load cell independent of the driver. The instrumental indentation with a Berkovich pyramidal indenter was used. The measurement was carried out at the maximum load of 200 mN and a loading rate of 400 mN/min.

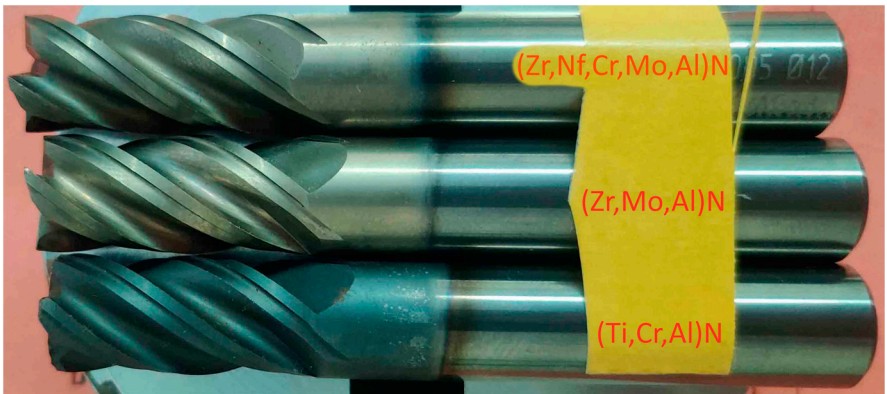

**Figure 1.** General view of the coated milling cutters (after milling).

The following equipment was used to study the coating structure:

- a transmission electron microscope (TEM) JEM 2100 (JEOL, Japan) at an accelerating voltage of 200 kV. The samples for the study (lamellas) were cut out using a Strata focused ion beam (FIB) 205 (FEI, USA),
- a scanning electron microscope (SEM) FEI Quanta 600 FEG.

The adhesion (molecular) component $f_{adh}$ of the coefficient of friction (COF) was determined using the method proposed by L. Sh. Shuster [68–72]. This method allows using the physical model of the cutting process, implemented on a special installation. A test coating was deposited on a carbide indenter with hemispherical ends. This indenter was placed between two flat inserts of Inconel 718 and compressed between these inserts with force N, simulating the process of cutting. The system consisting of the coated indenter and two inserts was heated to a predetermined temperature ranging from 0 to 900 °C. The indenter rotated around its own axis, and the force $F_{exp}$, required to continue the rotation, was measured. Based on the value of $F_{exp}$, the indenter radius $r_{ind}$, and the radius $R_{exp}$ of the indenter's trace on the inserts, the strength of the adhesive bonds per shear $\tau_n$ was determined as follows:

$$\tau_n = \frac{3}{4} \frac{F_{exp}}{\pi} \frac{R_{exp}}{r_{ind}^3}$$

Based on the value of the compressing force N applied to the indenter, the value of the normal stresses $p_r$ acting on the surface of the sphere was determined:

$$p_r = \frac{N}{n\, r_{ind}^2}$$

The adhesion (molecular) component $f_{adh}$ of the COF for each measurement temperature was determined as follows:

$$f_{adh} = \frac{\tau_n}{p_r}$$

## 3. Results

### 3.1. Hardness, Elastic Modulus, and Critical Fracture Load in Scratch Testing

A considerably high hardness was measured for all three coatings (see Table 2). The (Ti,Cr,Al)N and (Zr,Mo,Al)N coatings showed an almost identical hardness (given the spread of the values), and the (Zr,Hf,Cr,Mo,Al)N coating showed a slightly lower hardness. All three coatings demonstrated a critical fracture load sufficient for the operation during scratch testing (at least 38 N).

**Table 2.** Results of the study considering the mechanical properties of the coatings.

| Coating | Hardness, GPa | Elastic Modulus, GPa | Critical Fracture Load $L_{C2}$, N |
|---------|---------------|----------------------|------------------------------------|
| (Ti,Cr,Al)N | 31.9 ± 1.4 | 580.50 ± 22.4 | 38 |
| (Zr,Mo,Al)N | 32.3 ± 1.2 | 432.15 ± 21.4 | >40 |
| (Zr,Hf,Cr,Mo,Al)N | 27.3 ± 1.3 | 532.23 ± 26.1 | >40 |

### 3.2. Adhesion Component of the Coefficient of Friction (COF)

One of the key properties of wear-resistant coatings for cutting tools is the value of the COF. Figure 2a depicts the results of the studies considering the variation in the adhesion component $f_{adh}$ of the COF for the pair "coated indenter–inserts of Inconel 718" depending on temperature. When the temperature rose to 500 °C and higher, the value of $f_{adh}$ for the sample with the (Ti,Cr,Al)N coating began to increase noticeably, significantly exceeding the values of $f_{adh}$ for the samples with the (Zr,Mo,Al)N and (Zr,Hf,Cr,Mo,Al)N coatings. When the temperature reached 700 °C and rose further, the value of $f_{adh}$ for the sample with the (Ti,Cr,Al)N coating began to decrease, but still remained significantly higher in comparison with the values of $f_{adh}$ for the samples with the coatings consisting of (Zr,Mo,Al)N and (Zr,Hf,Cr,Mo,Al)N. With the rising temperature, there was a similar trend in the variation of $f_{adh}$ for the samples with the (Zr,Mo,Al)N and (Zr,Hf,Cr,Mo,Al)N coatings: $f_{adh}$ increased with a rise in temperature up to 700 °C and then decreased. However, these variations were much less pronounced in comparison with those observed for the (Ti,Cr,Al)N coating. As the variations in $f_{adh}$ for the coatings consisting of (Zr,Mo,Al)N and (Zr,Hf,Cr,Mo,Al)N were hardly distinguishable compared to the significant variation in $f_{adh}$ measured for the (Ti,Cr,Al)N coating, the variations in $f_{adh}$ for the coatings of (Zr,Mo,Al)N and (Zr,Hf,Cr,Mo,Al)N were considered separately and on a large scale (Figure 2b). At the temperature of 400 °C, the coating consisting of (Zr,Hf,Cr,Mo,Al)N had a slightly larger value of $f_{adh}$, but as the temperature rose, the differences in $f_{adh}$ between the coatings almost disappeared. Thus, in terms of the value of $f_{adh}$ at temperatures similar to those in the cutting zone, the coatings consisting of (Zr,Mo,Al)N and (Zr,Hf,Cr,Mo,Al)N appeared preferable to the (Ti,Cr,Al)N coating. However, these two coatings were very close in their tribological properties. It can be assumed that the introduction of Hf and Cr into the composition of the coating did not have a noticeable effect on the tribological properties.

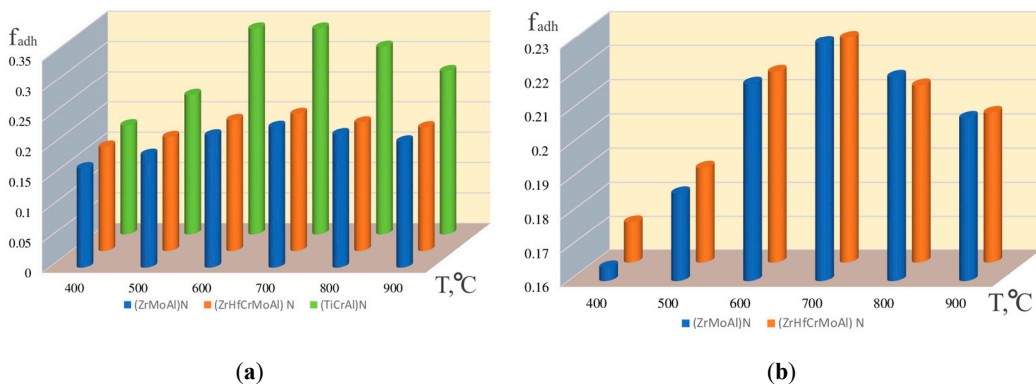

(**a**) (**b**)

**Figure 2.** Variation in the adhesion component $f_{adh}$ of the COF depending on the temperature. (**a**) Comparison of three coatings, (**b**) comparison of the (Zr,Mo,Al)N and (Zr,Hf,Cr,Mo,Al)N coatings only.

### 3.3. Structure and Phase Composition

The studies of the coating structure showed that the transition layer had a monolithic structure, and the wear-resistant layer had a nanolayer structure (Figure 3a). The structures

of the nanolayers were clearly distinguishable in the wear-resistant layer of all three coatings. The modulation period λ was 45, 48, and 50 nm for the coatings consisting of (Ti,Cr,Al)N, (Zr,Mo,Al)N, and (Zr,Hf,Cr,Mo,Al)N, respectively. Thus, it can be argued that the coatings under consideration had identical structures.

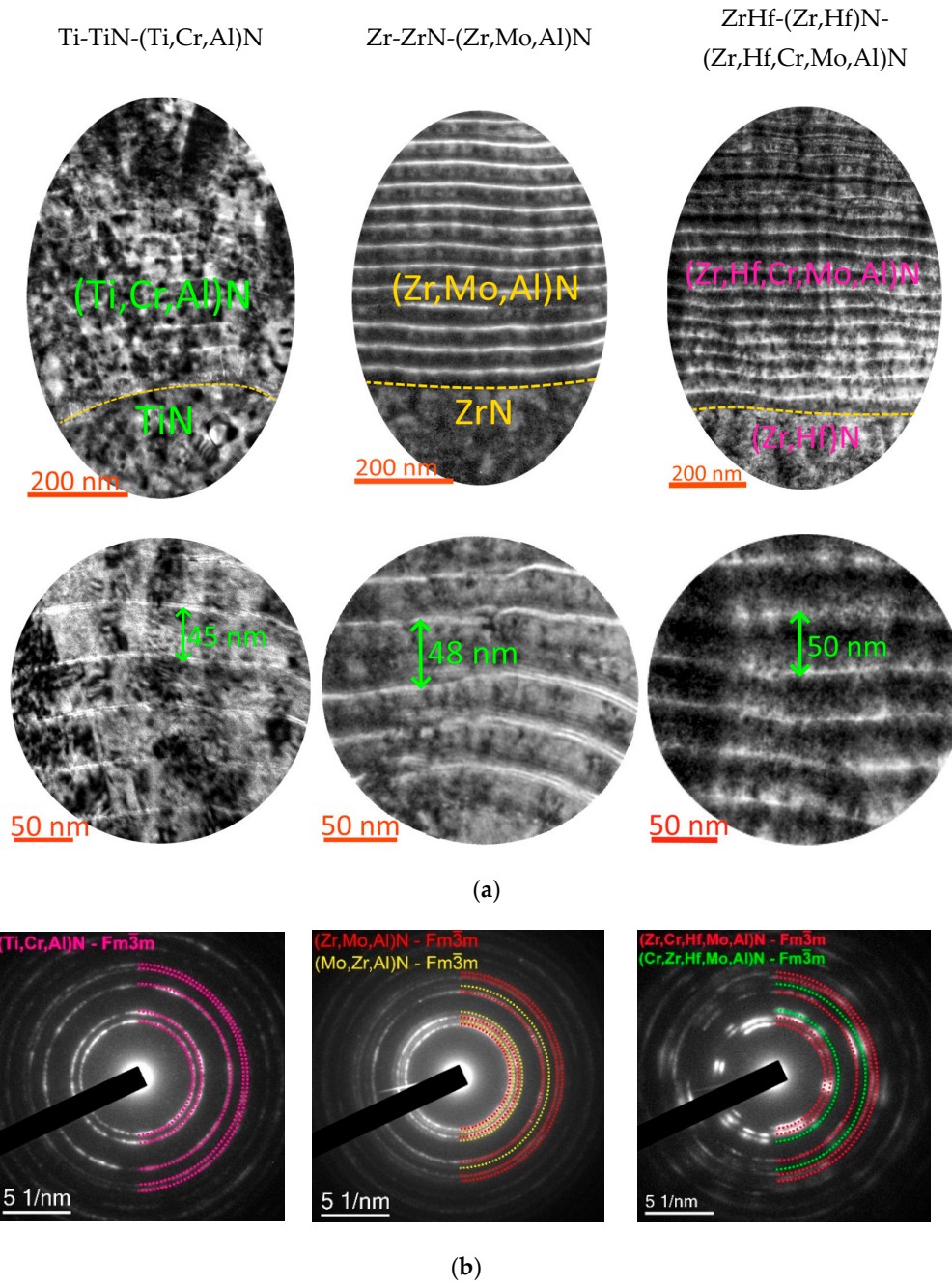

**Figure 3.** (**a**) Structure composition, (**b**) electron diffraction patterns of the coatings under consideration.

The phase composition of the coatings (their wear-resistant layer) was considered. The (Ti,Cr,Al)N coating showed only one cubic phase of c-(Ti,Cr,Al)N, which is a solid solution of Cr and Al in c-TiN. In the coating consisting of (Zr,Mo,Al)N, two cubic phases were detected, including c-(Zr,Mo,Al)N and c-(Mo,Zr,Al)N, formed on the basis of c-ZrN and c-MoN, respectively. The (Zr,Hf,Cr,Mo,Al)N coating also formed two cubic phases, including c-(Zr,Cr,Hf, Mo,Al)N and c-(Cr,Zr,Hf, Mo,Al)N, based on c-ZrN and c-CrN, respectively, in which the other coating elements were dissolved, forming a substitution solid solution.

Data on the presence of two cubic phases were also confirmed by X-ray diffraction (XRD). The coating consisting of (Ti,Cr,Al)N exhibited a typical columnar crystal structure, and the growth of the columnar grains was not always interrupted by the nanolayer boundaries. A similar, although less pronounced, pattern was observed for the (Zr,Hf,Cr,Mo,Al)N coating. On the contrary, in the (Zr,Mo,Al)N coating, columnar grains formed only within one modulation period.

### 3.4. Wear Resistance in the End Milling of Inconel 718

The experiments on the wear dynamics of end milling cutters with the considered coatings during the milling of the Inconel 718 alloy (Figure 4) demonstrated that all three coatings showed a noticeable increase in wear resistance (about two times that of an uncoated tool). The (Zr,Mo,Al)N coating provided maximum wear resistance, and the (Ti,Cr,Al)N-coated sample exhibited the highest wear rate. The tool coated with Zr,Hf,Cr,Mo,Al)N had an average wear rate among the samples considered.

The consideration of the flank wear patterns on the coated tools revealed certain differences (Figure 5). In particular, the tool with the (Ti,Cr,Al)N coating showed the most noticeable chipping of the cutting edge (Figure 5a). Some signs of chipping were also observed on the tools coated with (Zr,Hf,Cr,Mo,Al)N. Among these samples, the smallest signs of chipping were detected in the sample coated with (Zr,Mo,Al)N. On this sample, the wear boundary was quite even, without significant chipping and potholes (Figure 6). At the same time, on the sample with other coatings, the wear boundary was quite uneven, with chipping and potholes. The least distortion of the cutting edge shape was observed for the sample with the (Zr,Mo,Al)N coating (Figure 5b). The coating consisting of (Zr,Mo,Al)N was mostly preserved on the flank face of the milling cutter, while the samples coated with (Ti,Cr,Al)N and (Zr,Hf,Cr,Mo,Al)N bore the signs of wear not only directly in the cutting zone, but also on the rake face in general, which may be explained by the specifics of the movement of the cut material flow during the milling.

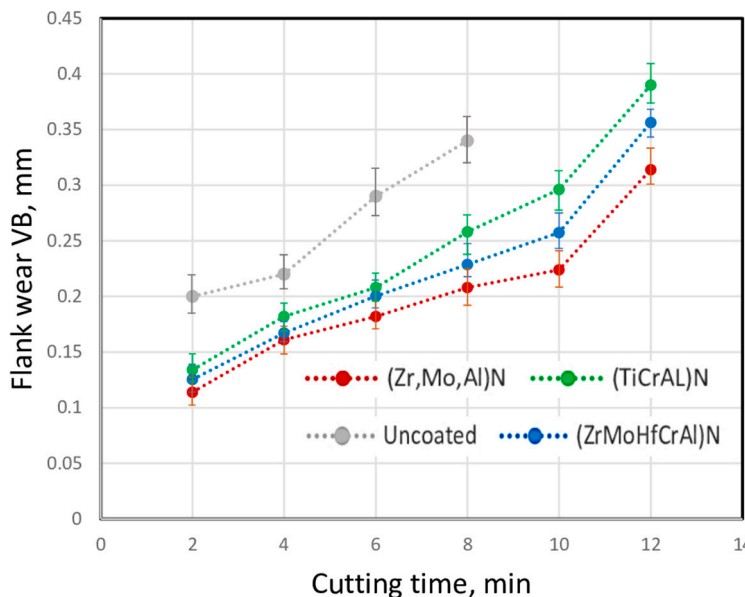

**Figure 4.** Wear dynamics o end milling cutters during the end milling of Inconel 718 alloy.

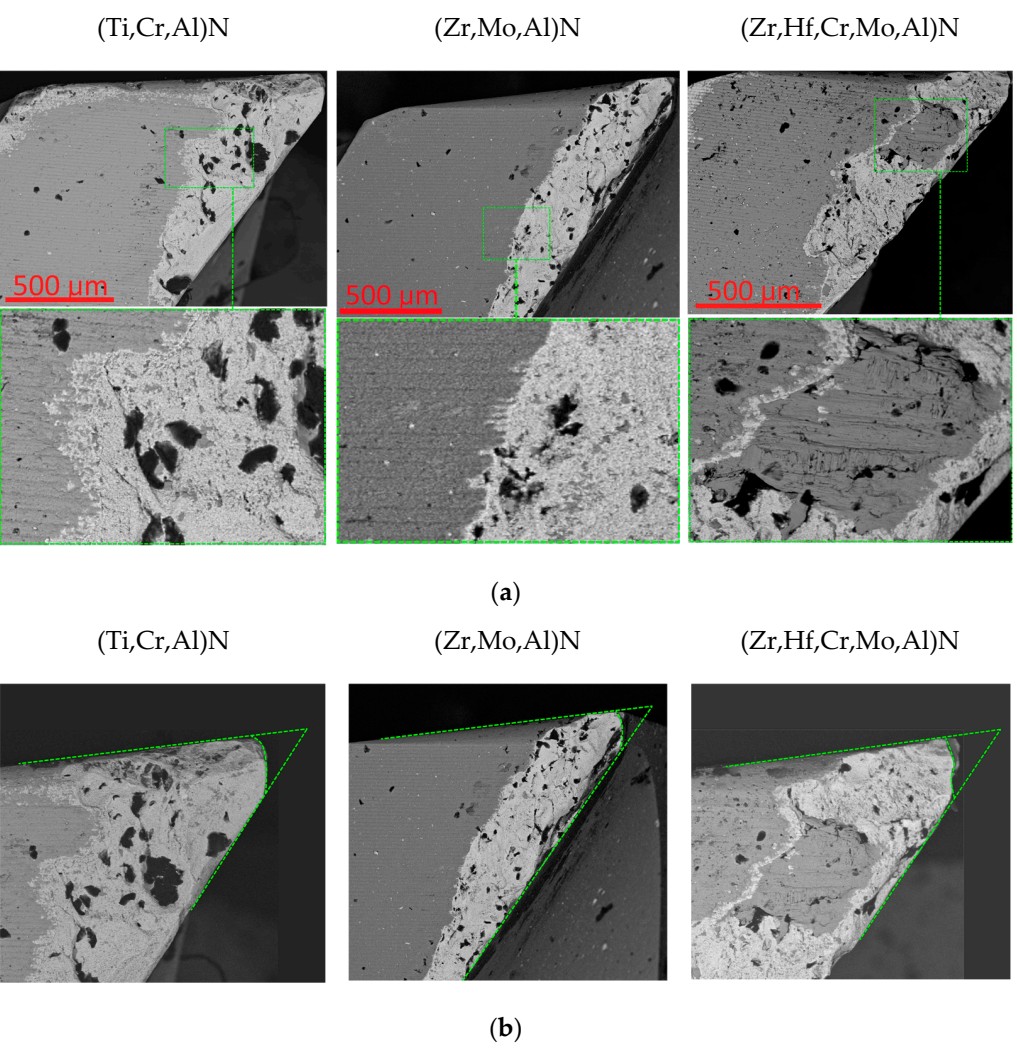

(**a**)

(**b**)

**Figure 5.** General pattern of wear on the tools with the considered coatings. (**a**) Flank wear pattern, (**b**) change in cutting edge geometry.

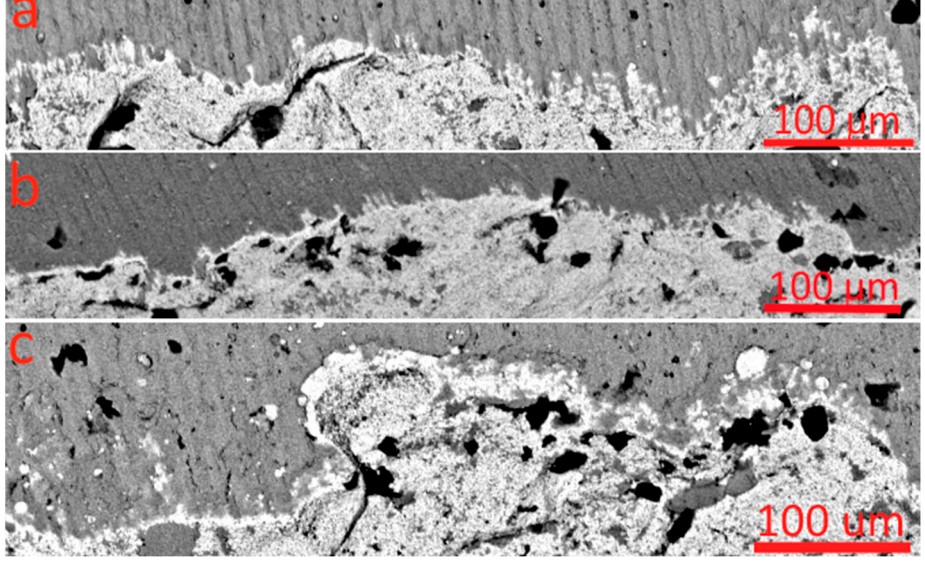

**Figure 6.** Boundary of the wear zone. (**a**) (Ti,Cr,Al)N, (**b**) (Zr,Mo,Al)N, (**c**) (Zr,Hf,Cr,Mo,Al)N.

The analysis of the map depicting the distribution of the main elements on the rake face of the worn milling cutters with the different coatings made it possible to select areas for cutting lamellas and to make cut sections (Figure 7).

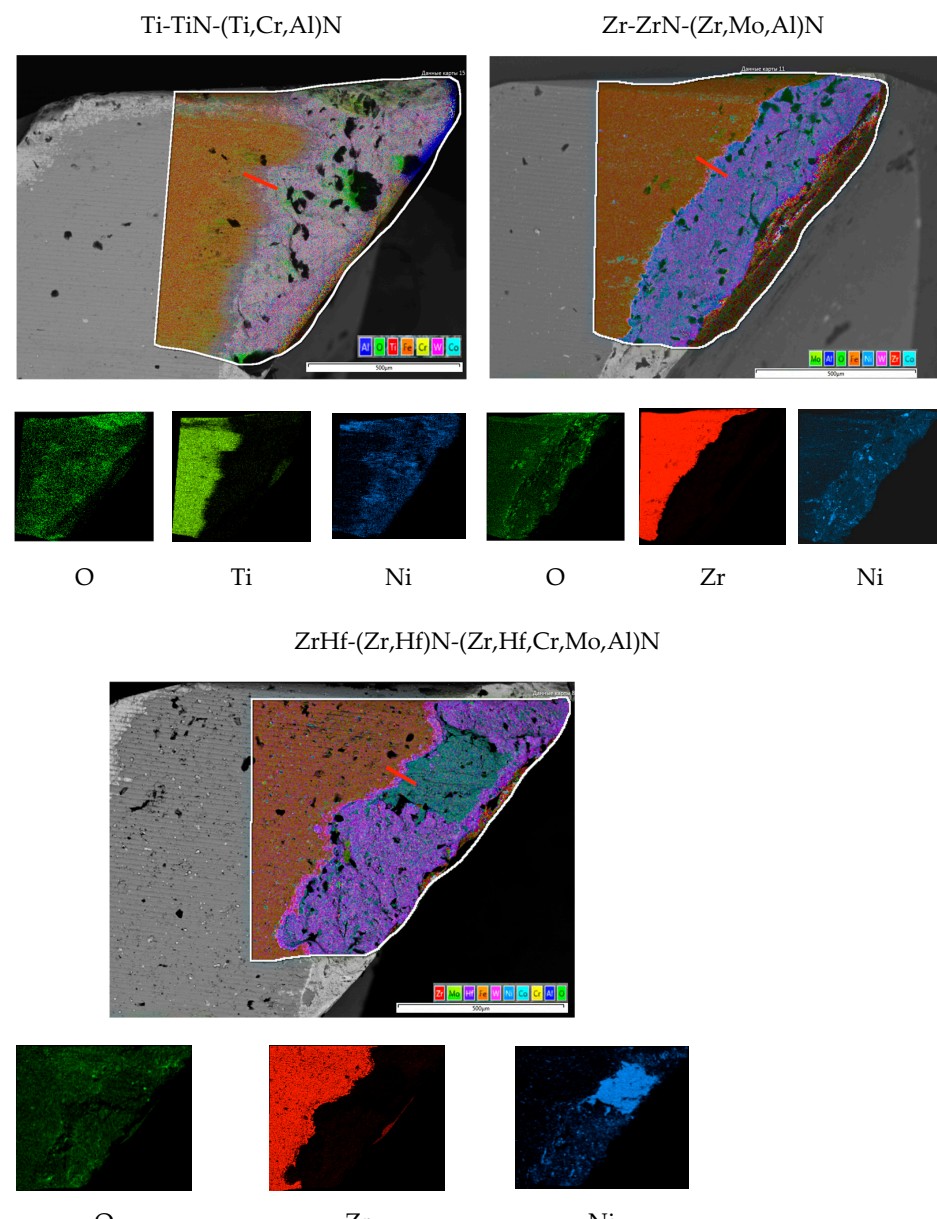

**Figure 7.** Mapping of the distribution of elements on the flank face of worn milling cutters with the considered coatings. The red line marks the places where the lamellas were cut out.

A separate consideration concerned the content of oxygen (for the analysis of oxidation processes), nickel (the main element of the material being machined), and the main elements of the coating (titanium or zirconium) for a more accurate determination of the wear boundary. The sample with the (Ti,Cr,Al)N coating exhibited a considerably active oxidation, and the intensity of the signal from oxygen was almost identical both on the surface of the coating and on the worn part of the tool, where no coating was retained. Based on the oxygen content, it was almost impossible to determine the coating wear boundary. Such active oxidation can be associated with an elevated temperature in the cutting zone, due to which both the oxidation of the coating and the oxidation of the carbide itself occurred. Given the considerably higher value of $f_{adh}$ for the (Ti,Cr,Al)N coating, a

greater heat release during the cutting process and, consequently, increased oxidation of both the coating and the worn surface of the carbide substrate can be assumed. For the sample with the (Zr,Mo,Al)N coating, the oxidation rate was noticeably lower. This sample bore localized spots with high oxygen content, which can be associated with the presence of adherents with a high content of iron (the only actively oxidizing element of the Inconel 718 alloy, see Table 1). The map of the distribution of oxygen on the (Zr,Hf,Cr,Mo,Al)N coating surface also showed its typical features. The coating wear boundary could be quite clearly determined here based on the distribution of oxygen. A noticeable increase in oxygen concentration was observed along this boundary. Thus, it can be assumed there were active oxidation processes in the coating at the wear boundary, and these processes could affect the cutting process in a certain way.

*3.5. Studies of the Mechanism of Wear, Diffusion, and Oxidation Processes in the Coating Consisting of (Ti,Cr,Al)N*

The mechanism of fracture of the (Ti,Cr,Al)N coating on the flank face of the milling cutter was considered during the milling of the Inconel 718 alloy. Figure 8a,b shows that the coating bore typical signs of brittle fracture at the boundary of the wear zone.

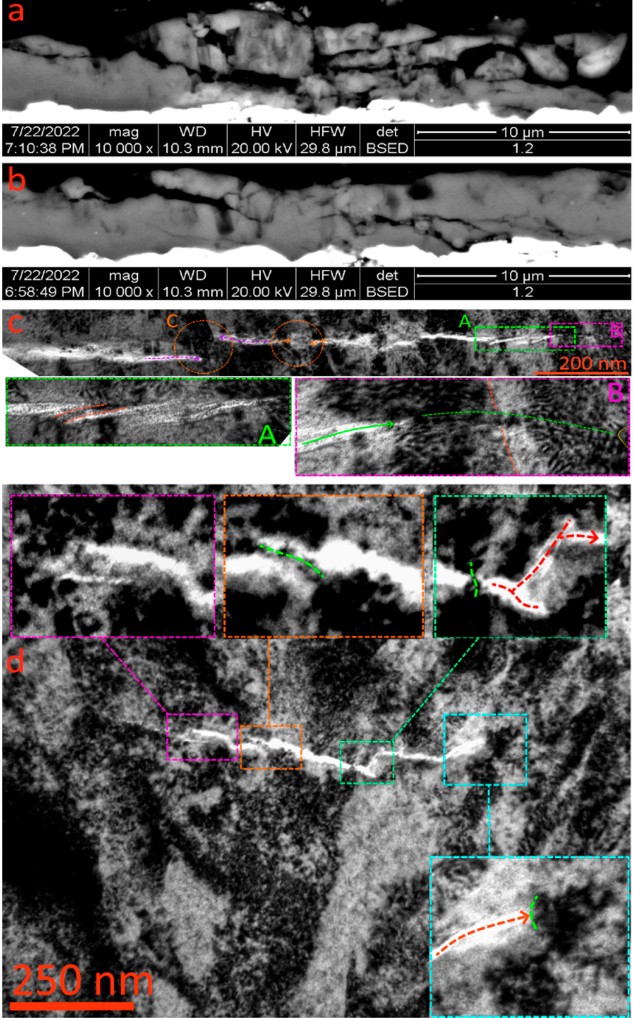

**Figure 8.** Study of the mechanism of fracture of the (Ti,Cr,Al)N coating on the flank face of the tool during milling. (**a,b**) General pattern of the coating fracture (SEM), (**c**) specific features of the longitudinal crack formation (TEM), (**d**) influence of the crystalline and nanolayer structure of the coating on crack propagation (TEM).

The formation of extended longitudinal cracks was accompanied with the chipping of large fragments of the coating. The grid of cracks cut through the nanolayer structure of the coating. Plastic bond bridges, inhibiting the propagation of the cracks, also formed between the nanolayers (Figure 8c,d). As noted above, the (Ti,Cr,Al)N coating formed columnar grains, and such structure developed over the entire thickness of the coating, passing through the interfaces between the nanolayers. The boundaries of such vertically directed grains could inhibit the development of horizontal cracks (Figure 8c) or change their direction (Figure 8d).

### 3.6. *Studies of the Mechanism of Wear, Diffusion, and Oxidation Processes in the Coating Consisting of (Zr,Mo,Al)N*

The mechanism of cracking in the (Zr,Mo,Al)N coating appeared significantly different. Despite inclined cracks and delaminations actively formed in the structure of the coating, in general, the coating looked much less worn in comparison with the (Ti,Cr,Al)N coating. Figure 9c shows how the crack propagation in the nanolayer structure of the coating slowed down. Along with the cracks, there were also delaminations between the coating nanolayers. As noted earlier, such delaminations play a positive role in terms of reducing the level of internal compressive stresses [73]. It can be noted that such delaminations can act as a kind of "plate springs," making the coating structure more elastic and less brittle.

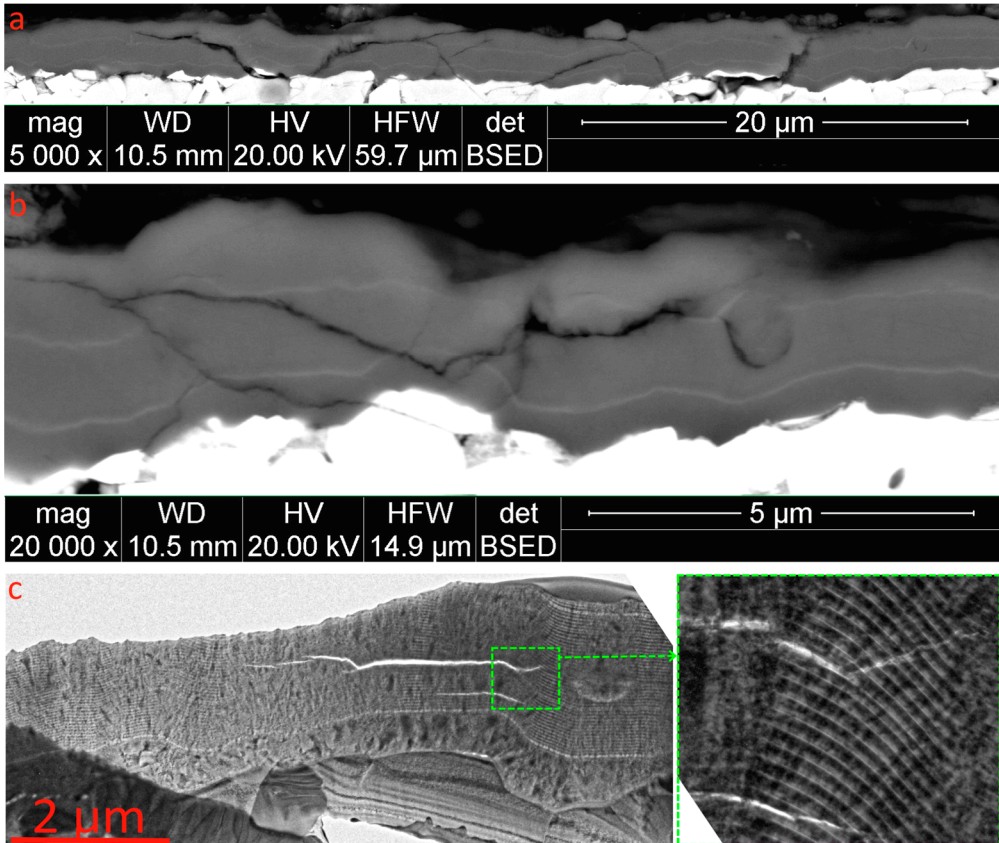

**Figure 9.** Study of the mechanism of fracture in the (Zr,Mo,Al)N coating on the flank face of the tool during milling. (**a**,**b**) General pattern of the coating fracture (SEM), (**c**) inhibition of crack propagation in the coating structure (TEM).

The more detailed study of the mechanisms of crack propagation in the structure of the (Zr,Mo,Al)N coating (Figure 10) showed that the development of cracks was inhibited by factors of the crystalline structure of the coating. In particular, the direction of crack propagation could change under the influence of interfaces between the nanolayers (Area D, Figure 10). In this case, a crack not only changed its direction, but also branched. Under

these transformations, the energy of crack propagation decreased, which ultimately slowed down the crack expansion [73]. In the conditions of the nanolayer structure, a crack slowed down, and a sequence of co-cracks formed, separated by plastic bond bridges (Area B, Figure 10). Individual strong grains also affected the crack propagation (Area C, Figure 10). Therefore, the (Zr,Mo,Al)N coating had higher resistance to cracking in comparison with the (Ti,Cr,Al)N coating. This property is of particular importance in intermittent cutting conditions, being affected by the action of alternating loads that provoke the propagation of cracks and the fatigue failure of the coating.

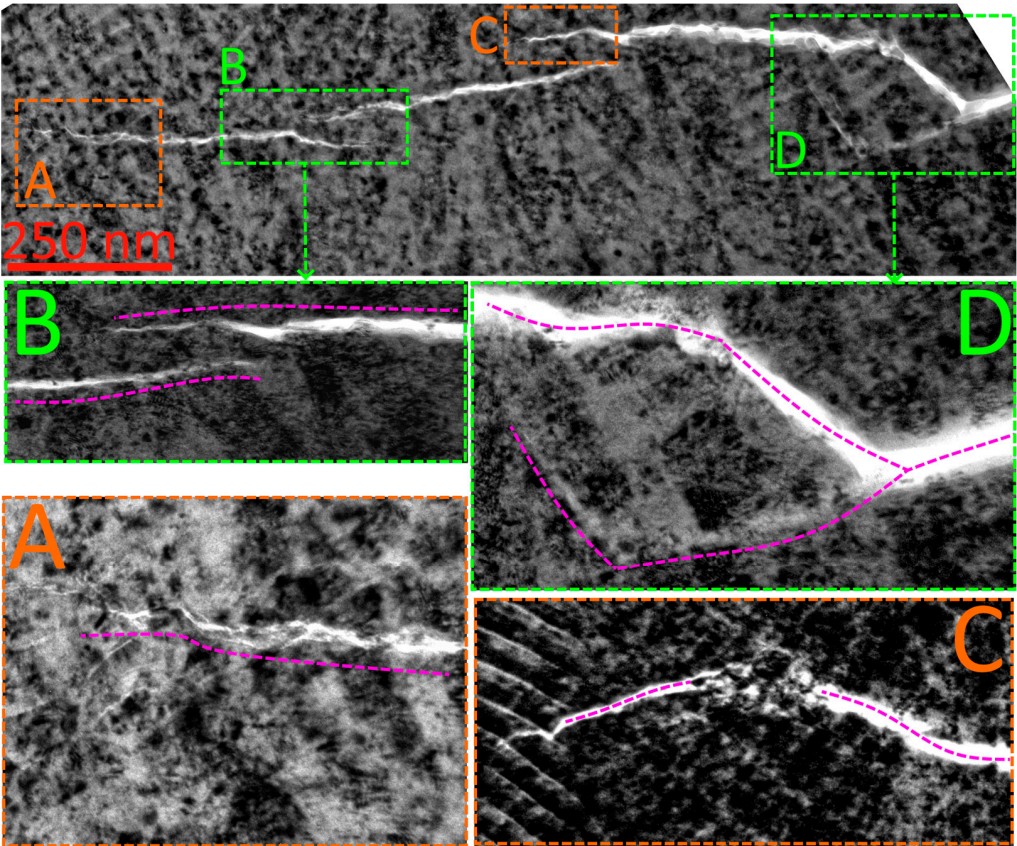

**Figure 10.** Factors affecting the propagation of cracks in the structure of the (Zr,Mo,Al)N coating.

*3.7. Studies of the Mechanism of Wear, Diffusion, and Oxidation Processes in the Coating Consisting of (Zr,Hf,Cr,Mo,Al)N*

The mechanism of wear typical for the coating consisting of (Zr,Hf,Cr,Mo,Al)N differed significantly from those observed for the (Ti,Cr,Al)N and (Zr,Mo,Al)N coatings considered above. While for the coatings considered earlier, cracking and brittle fracture were the main causes of fracture, for the (Zr,Hf,Cr,Mo,Al) coating, oxidation wear was a key factor in the same cutting conditions. No noticeable signs of oxidation wear were observed in the coatings considered above, but the coating consisting of (Zr,Hf,Cr,Mo,Al)N demonstrated the formation of a region with a significantly changed structure (Figure 11). This region exhibited a slight expansion of the nanolayers, and their structure appeared looser. The selected area electron diffraction (SAED) analysis of this region revealed the presence of a large amount of $(Zr,Hf)O_2$ oxide. Along with the oxide, certain amounts of the initial cubic phase of c-(Zr,Cr,Hf,Mo,Al)N and of the chromium nitride CrN were retained in this region. The presence of CrN can be explained by a more active decomposition of c-(Cr,Zr, Hf,Mo,Al)N, the second cubic phase of the coating, when free Zr and Hf were released from the solid solution, forming their corresponding oxides at elevated temperatures and in the presence of oxygen. The formation of Mo and Al oxides can also be assumed, but

their detection was complicated by their insignificant content in the coating in comparison with other elements. Therefore, it can be assumed that the c-(Zr,Cr,Hf,Mo,Al)N phase had a higher thermal stability in comparison with the c-(Zr,Cr,Hf,Mo,Al)N phase.

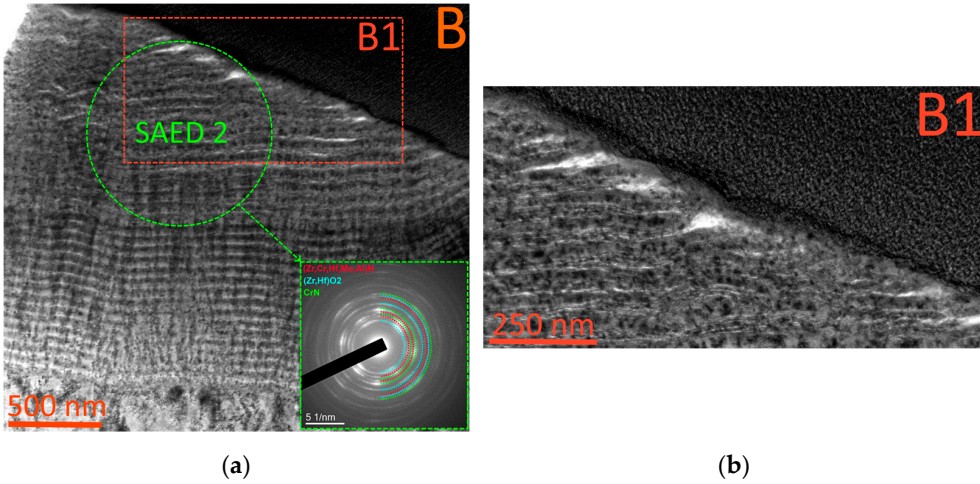

(**a**)　　　　　　　　　　　　　　　　　(**b**)

**Figure 11.** (**a**) Pattern of oxidation wear in the (Zr,Hf,Cr,Mo,Al)N coating. (**b**) Disruption of the nanostructure of the coating under the influence of oxidation.

The mechanism of cracking in the coating consisting of (Zr,Hf,Cr,Mo,Al)N was related to the oxidation processes. In general, no formation of any noticeable number of cracks was detected in this coating. The oxidized region exhibited multiple delaminations between the nanolayers (Figure 12a,b). There were also delaminations and longitudinal cracks in the non-oxidized region of the coating (Figure 12c,d). It should be noted that in the (Zr,Hf,Cr,Mo,Al)N coating, there were numerous plastic bond bridges between the nanolayers which acted as sufficiently effective inhibitors of the cracks (Figure 12d).

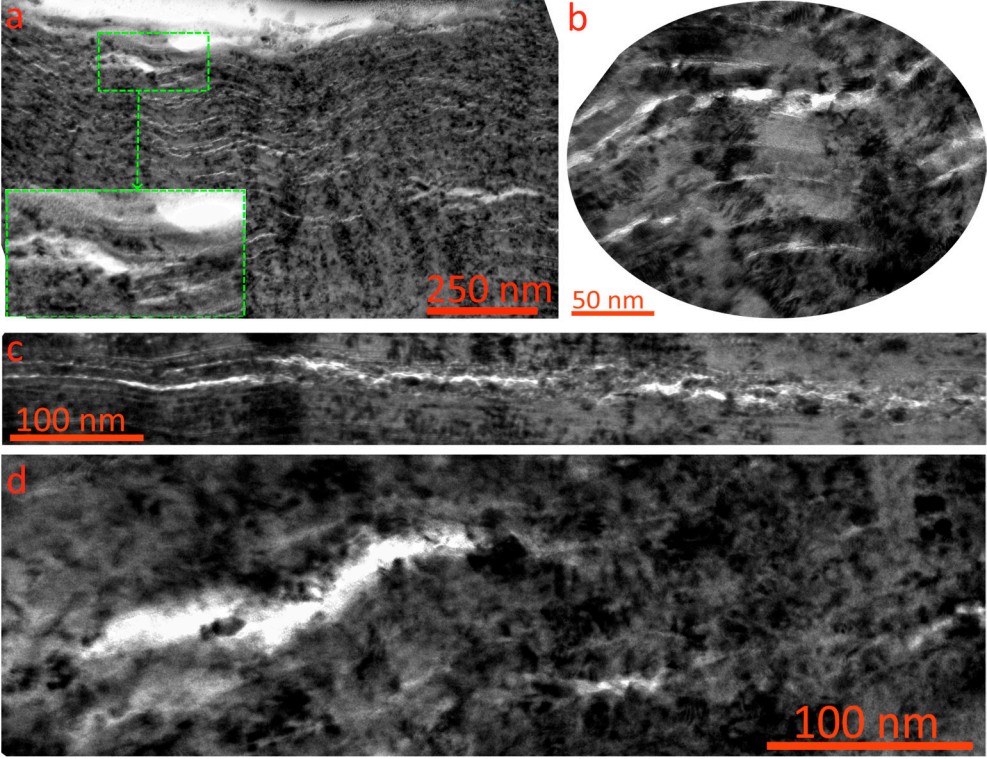

**Figure 12.** (**a–d**) Pattern of cracking in the (Zr,Hf,Cr,Mo,Al)N coating.

## 4. Conclusions

Three coatings consisting of Ti-TiN-(Ti,Cr,Al)N, Zr-ZrN-(Zr,Mo,Al)N, and ZrHf-(Zr,Hf)N-(Zr,Hf,Cr,Mo,Al)N with identical thickness and nanostructure parameters were considered. Based on the results of the conducted studies, the following conclusions can be drawn:

- The Ti-TiN-(Ti,Cr,Al)N and Zr-ZrN-(Zr,Mo,Al)N coatings have close values of hardness (31–32 GPa), while the hardness of the ZrHf-(Zr,Hf)N-(Zr,Hf,Cr,Mo,Al)N coating is slightly lower (about 27 GPa).
- The Zr-ZrN-(Zr,Mo,Al)N and ZrHf-(Zr,Hf)N-(Zr,Hf,Cr,Mo,Al)N coatings have noticeably lower values of the adhesion component $f_{adh}$ of the COF at elevated temperatures in comparison with the Ti-TiN-(Ti,Cr,Al)N coating.
- All three considered coatings provided an increase in the wear resistance of end milling cutters by about 2 times compared to uncoated tools. At the same time, the Zr-ZrN-(Zr,Mo,Al)N-coated tool demonstrated the highest wear resistance among all samples.
- We found significant differences in the mechanisms of wear for the tools with the examined coatings. For the coatings consisting of Ti-TiN-(Ti,Cr,Al)N and Zr-ZrN-(Zr,Mo,Al)N, active cracking was typical, while for the coating consisting of ZrHf-(Zr,Hf)N-(Zr,Hf,Cr,Mo,Al)N, the oxidation process was a key factor. The Zr-ZrN-(Zr,Mo,Al)N coating exhibitsd better resistance to cracking in comparison with the Ti-TiN-(Ti,Cr,Al)N coating.

Thus, the Zr-ZrN-(Zr,Mo,Al)N coating demonstrated high hardness and a low value of the adhesion component $f_{adh}$ of the COF at elevated temperatures. At the same time, this coating demonstrated good resistance to cracking and oxidation in the conditions of nickel alloy milling. Based on the above, the Zr-ZrN-(Zr,Mo,Al)N coating can be considered a good choice as a wear-resistant coating for the end milling of the Inconel 718 alloy.

**Author Contributions:** Conceptualization, A.V.; methodology, A.V. and F.M.; validation, A.M., M.M. (Mars Migranov) and A.T.; investigation, F.M., N.A. and M.M. (Mars Migranov); resources, I.A.; data curation, M.M. (Maxim Mikhailov); writing—original draft preparation, A.V.; project administration, I.A.; funding acquisition, I.A. All authors have read and agreed to the published version of the manuscript.

**Funding:** The results of this work were obtained as part of the work under the Agreement on the provision of subsidies dated December 14, 2020 No. 075-11-2020-032 (state contract identifier—000000S207520RNU0002) with the Ministry of Science and Higher Education of the Russian Federation.

**Data Availability Statement:** Data available on request due to privacy restrictions or ethical reasons.

**Conflicts of Interest:** The authors declare no conflict of interest.

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
