# Peer review of "Specific Application Features of Ti-TiN-(Ti,Cr,Al)N, Zr-ZrN-(Zr,Mo,Al)N, and ZrHf-(Zr,Hf)N-(Zr,Hf,Cr,Mo,Al)N Multilayered Nanocomposite Coatings in End Milling of the Inconel 718 Nickel-Chromium Alloy"

_jcs, doi:10.3390/jcs6120382_

Round 1

Reviewer 1 Report

Report on the submission: specific characteristics of the use of Ti-TiN-(Ti,Cr,Al)N, Zr-ZrN-(Zr,Mo,Al)N, and ZrHf- (Zr,Hf)N-(Zr,Hf,Cr,Mo,Al)N multilayered nanocomposite coatings in final milling of Inconel 718 nickel-chromium alloy

The paper is well written, the introduction introduces the topic well, and the methods used are appropriate for this study. The results, the associated discussion, and the conclusions drawn from them make sense to me. I would therefore recommend publishing this paper in the Journal of Composites Science after the following questions/comments have been answered by the authors (minor revision):

- Please add in the experimental part the chamber pressure during the coatings, as well as the substrate temperature

- An indentation force of 200 mN seems too high for hardness measurement of such thin PVD layers. Please indicate the measured indentation depth and comment and discuss its possible influence on the measurement results. As a basic rule (Bückle's rule), the penetration depth should be max. 1/10 of the layer thickness.
Furthermore, the presented results of the E-module are not credible, I would like to ask you to repeat these measurements. An elastic modulus greater than diamond for coatings that typically have moduli of about 400-450 GPa does not seem to be correctly evaluated.

- Please describe precisely how you account for forming two phases in the (Zr,Mo,Al)N and (Zr,Hf,Cr,Mo,Al)N layers. In my opinion, it is not sufficient to assume this only based on the shown SAEDs. Here, you would have to deliver more definitive proof (e.g., EDS measurements) to keep this statement.

Author Response

Author's Reply to the Review Report (Reviewer 1)

Report on the submission: specific characteristics of the use of Ti-TiN-(Ti,Cr,Al)N, Zr-ZrN-(Zr,Mo,Al)N, and ZrHf- (Zr,Hf)N-(Zr,Hf,Cr,Mo,Al)N multilayered nanocomposite coatings in final milling of Inconel 718 nickel-chromium alloy

The paper is well written, the introduction introduces the topic well, and the methods used are appropriate for this study. The results, the associated discussion, and the conclusions drawn from them make sense to me. I would therefore recommend publishing this paper in the Journal of Composites Science after the following questions/comments have been answered by the authors (minor revision):

The authors are grateful to the Reviewer for the high appreciation of their work and valuable recommendations for improving the quality of the manuscript.

- Please add in the experimental part the chamber pressure during the coatings, as well as the substrate temperature

Information added

- An indentation force of 200 mN seems too high for hardness measurement of such thin PVD layers. Please indicate the measured indentation depth and comment and discuss its possible influence on the measurement results. As a basic rule (Bückle's rule), the penetration depth should be max. 1/10 of the layer thickness.

The thickness of the investigated coatings is in the range of 3-5 µm. Indentation was carried out on a mechanical tester SV-500 Nanovea with a Berkovich indenter in accordance with ISO/FDIS 14577-2015, which is a trihedral pyramid with an angle of 65.03 degrees between the axis and the face. The indentation force is always selected during indentation, i.e. at different forces, the loading / unloading curve is analyzed and selected taking into account the characteristics of the coating, depending on the physical and mechanical properties (hardness, viscosity, etc.). The penetration depth of the indenter is in the range of 0.45...0.5 µm, which formally does not contradict the Bückle's rule.

Furthermore, the presented results of the E-module are not credible, I would like to ask you to repeat these measurements. An elastic modulus greater than diamond for coatings that typically have moduli of about 400-450 GPa does not seem to be correctly evaluated.

The authors apologize. Since we were using newly acquired equipment (and hence measurement methodology), some data was not correctly processed. A large number of measurements (25) were carried out, after which the average value was calculated. Among the values ​​there were obviously some erroneous data (noticeably larger than 1000 GPa), which influenced the final results. We have now discarded these misleading data and recalculated. As a result, values ​​were obtained in the range from 432.15 ± 21.4 to 580.50 ± 22.4 GPa, which looks adequate. At the same time, the hardness data are much more homogeneous and, in our opinion, correspond to the real values.

 - Please describe precisely how you account for forming two phases in the (Zr,Mo,Al)N and (Zr,Hf,Cr,Mo,Al)N layers. In my opinion, it is not sufficient to assume this only based on the shown SAEDs. Here, you would have to deliver more definitive proof (e.g., EDS measurements) to keep this statement.

The phase composition was also studied by the X-ray diffraction (XRD) method. This method is less accurate for thin film analysis than the selected area electron diffraction pattern (SAED) method. However, it also gives a similar result, showing the presence of two cubic phases: (Zr,Hf,Cr,Mo,Al)N (based on ZrN) and (Cr,Zr,Hf,Mo,Al)N (based on CrN). In this case, only one phase is observed in the (Ti,Cr,Al)N layer (see below). An indication of the results of X-ray diffraction has been added to the manuscript.

Ti-TiN-(Ti,Cr,Al)N

Zr,Hf-(Zr,Hf)N-(Zr,Hf,Cr,Mo,Al)N

Reviewer 2 Report

The manuscript is well organized. however, this has to be checked for English language for being accepted. 

Author Response

Author's Reply to the Review Report (Reviewer 2)

The manuscript is well organized. however, this has to be checked for English language for being accepted. 

The authors are grateful to the Reviewer for the high appreciation of their work and valuable recommendations for improving the quality of the manuscript.

The authors conducted an additional check of the English language and made the necessary changes.

Reviewer 3 Report

The manuscript by Alexey et al reports the nanocomposite coatings with three different composition in end milling of Inconel 718 nickel-chromium alloy.  The manuscript is well-organized with microstructure-property correlation. I have two comments below.

1. How the coating depositing condition influence the microstructure and property? Are the results repeatable?

2. How is the hardness and resistance to cracking and oxidation of these coating compared to other commonly used or commercial coatings, as well as cubic-phase ZrN and CrN with less metal elements? 

3. Authors attributed the cracking of the coating to the oxidation. How high would the temperature be during these test? If the process is protected from air (such as in vacuum), will the cracks still happen?

Author Response

Author's Reply to the Review Report (Reviewer 3)

The manuscript by Alexey et al reports the nanocomposite coatings with three different composition in end milling of Inconel 718 nickel-chromium alloy.  The manuscript is well-organized with microstructure-property correlation. I have two comments below.

The authors are grateful to the Reviewer for the high appreciation of their work and valuable recommendations for improving the quality of the manuscript.

  1. How the coating depositing condition influence the microstructure and property?

Since our scientific team has been working in the field of coating deposition for many years, we used our experience to select the optimal deposition conditions. In particular, for each cathode, the arc current was chosen, determined for this metal as a result of a large number of experiments carried out since the 1990s. At the same time, of course, this issue is not closed and additional experiments on changing the process parameters can be justified. In particular, of interest is such a parameter as the rotation speed of the turntable, which determines the value of the coating modulation period (such studies are carried out, including by our scientific team - see, for example, doi:10.1016/j.surfcoat.2020.125402). Nitrogen pressure is also an important parameter. Despite the large amount of research in this area, further research may also be of interest.

Are the results repeatable?

Yes, definitely. By using the set process parameters, a coating with the same properties will be obtained. In this case, it should be taken into account that the spatial arrangement of the sample in the installation chamber affects the parameters of the coating structure (see doi.org/10.1016/j.vacuum.2022.111144).

  1. How is the hardness and resistance to cracking and oxidation of these coating compared to other commonly used or commercial coatings, as well as cubic-phase ZrN and CrN with less metal elements? 

Within the framework of this work, no direct comparison with ZrN and CrN coatings was carried out. The hardness of ZrN and CrN (having fairly close values) has been repeatedly determined and, according to various data, is 25.2 GPa (10.1016/j.surfcoat.2011.10.029) 28 GPa (10.1016/j.surfcoat.2012.03.002) or 31.8 GPa (doi:10.3365 /met.mat.2008.08.465). As can be seen, there is a significant difference in this value for different authors. It can be seen that, in terms of hardness (measured at room temperature), the ZrN and CrN coatings do not differ significantly from the multicomponent coatings we are considering. It is worth considering the fact that with increasing temperature there is a significant change (usually a decrease, but sometimes also an increase) in hardness. Since today's cutting conditions require high temperatures in the cutting zone (up to 1000 degrees Celsius or even higher), the heat resistance of tool materials in general and coatings in particular becomes a key issue. From this point of view, the introduction of additional elements into the coating composition (in particular, aluminum and molybdenum) makes it possible to increase the heat resistance of the coating, as well as to improve the antioxidant properties due to the formation of dense oxide films. This thesis has been repeatedly confirmed in a number of studies. From the point of view of practice, previous studies (including by our research team) show a noticeable advantage of coatings containing aluminum and some other metals (molybdenum, yttrium, hafnium, etc.) over coatings of a simple composition at high cutting speeds ( and, accordingly, high temperatures in the cutting zone). Based on this, we considered it correct to choose as an object of comparison not a two-component coating (for example, ZrN and CrN ), but a three-component coating based on the (Ti,Cr,Al)N system, including aluminum. Perhaps a ZrN-based coating (eg (Zr,Cr,Al)N) should have also been considered, but as work continues in this direction, we plan to do so in the future.

  1. Authors attributed the cracking of the coating to the oxidation. How high would the temperature be during these test?

The study was carried out under conditions of real cutting (milling) of the Inconel 718 alloy. It is difficult to accurately determine the temperature in the cutting zone, but the pyrometer data show values ​​of 800-900 degrees Celsius.

If the process is protected from air (such as in vacuum), will the cracks still happen?

Since real cutting was used, it is very difficult to conduct a study in a vacuum (for this it is necessary to isolate the entire milling machine). Sometimes an inert gas atmosphere is used - precisely to reduce oxidation. However, the use of such equipment is very expensive and is not justified economically (except in some cases when particularly high accuracy is required or metals such as magnesium are processed and an oxygen-containing environment is unacceptable).

Round 2

Reviewer 1 Report

The authors responded appropriately to the existing criticisms and made appropriate changes to their manuscript. I would therefore like to make a recommendation for the publication of the article. 

Reviewer 2 Report

The manuscript is now  acceptable in present form.

Reviewer 3 Report

Based on the revised manuscript, I recommend the publication.